# Is Involvement in Food Tasks Associated with Psychosocial Health in Adolescents? The EHDLA Study

**DOI:** 10.3390/nu17142273

**Published:** 2025-07-09

**Authors:** Mónica E. Castillo-Miñaca, María José Mendoza-Gordillo, Marysol Ruilova, Rodrigo Yáñez-Sepúlveda, Héctor Gutiérrez-Espinoza, Jorge Olivares-Arancibia, Susana Andrade, Angélica Ochoa-Avilés, Pedro Juan Tárraga-López, José Francisco López-Gil

**Affiliations:** 1School of Medicine, Universidad Espíritu Santo, Samborondón 092301, Ecuador; monelcami@hotmail.com (M.E.C.-M.); josefranciscolopezgil@gmail.com (J.F.L.-G.); 2United Nations Children’s Fund (UNICEF), Quito 170515, Ecuador; mamendoza@unicef.org (M.J.M.-G.); mruilova@unicef.org (M.R.); 3Centro de Investigación para la Salud en América Latina, Pontificia Universidad Católica del Ecuador, Quito 170530, Ecuador; 4Faculty Education and Social Sciences, Universidad Andres Bello, Viña del Mar 2260000, Chile; 5Faculty of Education, Universidad Autónoma de Chile, Santiago 7500912, Chile; hector.gutierrez@uautonoma.cl; 6AFySE Group, Research in Physical Activity and School Health, School of Physical Education, Faculty of Education, Universidad de las Américas, Santiago 8370040, Chile; jolivares@udla.cl; 7Grupo de Investigación Alimentación, Nutrición, Salud y Actividad Física, Departamento de Biociencias, Facultad de Ciencias Químicas, Universidad de Cuenca, Cuenca 010201, Ecuador; susana.andrade@ucuenca.edu.ec (S.A.); angelica.ochoa@ucuenca.edu.ec (A.O.-A.); 8Faculty of Medicine, University of Castilla la Mancha, 02071 Albacete, Spain; 9Gerencia de Atención Integrada de Albacete, Servicio de Salud de Castilla-La Mancha, 45006 Albacete, Spain; 10Vicerrectoría de Investigación y Postgrado, Universidad de Los Lagos, Osorno 5311269, Chile

**Keywords:** adolescence, mental health, psychological aspects, youths, healthy eating behavior, psychosocial development

## Abstract

**Background**: While some evidence supports the benefits of food-related tasks, research examining their association with psychosocial health in adolescents remains scarce. The aim of this study was to examine the association between Spanish adolescents’ involvement in food-related household tasks and their psychosocial health. **Methods:** This cross-sectional study used secondary data from the original Eating Healthy and Daily Life Activities (EHDLA) study. The final sample comprised 273 boys (43.0%) and 361 girls (57.0%). Adolescents self-reported their weekly frequency of involvement in two food-related tasks: meal preparation and grocery shopping, with responses ranging from ‘never’ to ‘seven times’. Psychosocial health was assessed using the 25-item self-report version of the Strengths and Difficulties Questionnaire (SDQ), comprising five subscales: emotional problems, conduct problems, hyperactivity, peer problems, and prosocial behavior. A total difficulties score was calculated by summing the first four subscales. Generalized linear models were used to evaluate associations between the frequency of food task involvement (categorized into five levels) and SDQ outcomes. All models were adjusted for age, sex, socioeconomic status, body mass index, sleep duration, physical activity, sedentary behavior, and energy intake. **Results:** Concerning to the frequency of helping to prepare food for dinner, an inverse association was observed between food preparation involvement and several psychosocial problems. Adolescents who helped seven times per week reported significantly lower scores in conduct problems (*B* = −2.00; 95% CI −3.30 to −0.69; *p* = 0.003), peer problems (*B* = −2.83; 95% CI −4.29 to −1.38; *p* < 0.001), internalizing problems (*B* = −3.90; 95% CI −7.03 to −0.77; *p* = 0.015), and total psychosocial difficulties (*B* = −5.74; 95% CI −10.68 to −0.80; *p* = 0.023), compared to those who never helped. Conversely, those who helped seven times per week had higher prosocial behavior than their counterparts who never helped (*B* = 1.69; 95% CI: 0.14 to 3.24; *p* = 0.033). Regarding the frequency of helping to shop for food, similar patterns were found, with lower conduct problems (*B* = −2.11; 95% CI −3.42 to −0.81; *p* = 0.002), peer problems (*B* = −2.88; 95% CI −4.34 to −1.42; *p* < 0.001), internalizing problems (*B* = −4.16; 95% CI −7.28 to −1.04; *p* = 0.009), and total psychosocial difficulties (*B* = −6.31; 95% CI −11.24 to −1.39; *p* = 0.012) associated with more frequent involvement, especially among those who helped five or more times per week. Conversely, adolescents who helped seven times per week had higher prosocial behavior than their peers who never helped (*B* = 1.56; 95% CI: 0.01 to 3.11; *p* = 0.049). **Conclusions:** Although adolescent psychosocial health is influenced by multiple factors, our findings suggest that regular involvement in food-related household tasks may serve as a protective factor against conduct problems, peer problems, internalizing problems, and total difficulties, while also enhancing prosocial behavior. However, given the cross-sectional design, conclusions regarding causality should be made cautiously, and further longitudinal research is needed to confirm these associations and assess their long-term impact. These results highlight the relevance of daily structured routines, such as meal preparation and grocery shopping, as potential support for mental well-being during adolescence.

## 1. Introduction

Psychosocial health problems continue to affect a significant proportion of the world’s population, including adolescents [1] Adolescence is a period of significant development characterized by profound changes in biological, cognitive, psychosocial, and emotional domains [2]. This period is associated with a high prevalence of psychosocial health disorders [3]. Research indicates that between 10% and 20% of adolescents face some type of psychosocial health issue; however, most of these cases remain unrecognized and untreated [4]. Problems such as anxiety, depression, hyperactivity, and difficulties in peer interactions can negatively impact young people’s personal growth, academic performance, and overall well-being over time [5,6]. Furthermore, adolescents’ involvement in food-related household tasks is often shaped by gender norms, which can influence both the frequency of participation and the perceived meaning of these responsibilities [7,8]. For example, in Spain and other European countries, traditional gender roles may lead girls to engage more frequently in domestic tasks, while boys may be less encouraged to participate [9,10]. These dynamics could have implications for both household contributions and psychosocial outcomes, given the potential of these tasks to foster self-esteem, autonomy, and social skills [11].

Lifestyle factors, such as physical activity and diet, play a key role in the mental health of adolescents [12,13]. A balanced diet can promote emotional and social well-being, whereas unhealthy eating patterns increase the risk of emotional and behavioral disorders [14]. In addition to diet quality, food-related tasks, such as cooking and grocery shopping, not only promote autonomy and decision making [15] but also contribute to emotional regulation, problem solving, and family cohesion [16]. In this context, it is important to consider that food and mealtime hold strong cultural and emotional significance in Spain, where shared meals are seen as key opportunities for family interaction and cohesion [17]. Adolescents’ involvement in food preparation may therefore represent not only practical skill development but also participation in meaningful social and emotional rituals that support psychosocial well-being [18]. Similarly, studies have indicated that involvement in these activities could promote family interaction and strengthen communication between parents and children [19].

Previous research has suggested that structured routines, including cooking and grocery shopping, may contribute to emotional well-being and psychosocial development by promoting self-regulation, problem-solving skills, and social interaction [20,21]. Cooking has been associated with neurobiological activation through motor coordination, which may influence stress regulation and emotional stability [22,23]. Additionally, food-related tasks involve life skills, such as planning and organization, which the World Health Organization identifies as essential for psychosocial competence [24]. It is also necessary to acknowledge the growing influence of digital and social media on adolescents’ food-related behaviors [25]. Exposure to cooking tutorials, influencer content, and food-related trends on platforms like Instagram or TikTok can shape adolescents’ interest and attitudes toward food preparation [25]. These influences may either enhance motivation or promote unrealistic standards that affect mental well-being [26].

Although direct evidence on the relationship between food task involvement and psychosocial health in adolescents remains limited, some studies suggest potential benefits. For example, interventions promoting culinary nutrition education among children have shown improvements in self-efficacy, food-related confidence, and family food environments [27]. Other research indicates that psychosocial factors such as autonomy and home food availability influence adolescents’ involvement in healthy meal preparation [28].

While some evidence supports the benefits of food-related tasks [15,29,30], research examining their association with psychosocial health in adolescents remains scarce. Most studies have focused on general dietary patterns rather than their specific role in adolescent psychosocial health [31,32,33,34]. Further research is needed to determine whether participation in food preparation and grocery shopping is associated with emotional and behavioral difficulties. It should also be noted that the majority of studies exploring these associations are observational, limiting the ability to infer causality, although a few intervention studies exist. Therefore, this study investigates whether regular involvement in food-related household tasks, such as meal preparation and grocery shopping, is associated with fewer emotional and behavioral difficulties and greater prosocial behavior in Spanish adolescents, hypothesizing that such engagement may support their psychosocial well-being.

## 2. Materials and Methods

### 2.1. Study Design and Population

This study was based on the Eating Healthy and Daily Life Activities (EHDLA) project, which included a representative sample of adolescents from the *Valle de Ricote* (Region of Murcia, Spain) during the 2021–2022 academic year. The methodological framework of the EHDLA project has been described in a prior publication [35]. The sample was selected from the three educational centers in the region, with adolescents aged between 12 and 17 years. Of the initial 1378 participants, 640 (46.4%) were excluded due to missing data on the SDQ total difficulties score, resulting in 738 participants (53.6%) eligible for analysis. Then, 38 participants (2.8%) were excluded due to missing data on involvement in household food tasks, yielding 700 participants (50.8%). Subsequently, 39 participants (2.8%) were excluded because of missing body mass index data, resulting in 661 participants (48.0%). Another 11 participants (0.8%) were excluded due to missing physical activity data (i.e., YAP-S), leaving 650 participants (47.2%). Finally, 16 participants (1.2%) were excluded because of missing data on energy intake, yielding a final analytical sample of 634 adolescents (46.0%). The final sample comprised 273 boys (43.0%) and 361 girls (57.0%). Participation in the study required informed consent from parents or legal guardians, as well as assent from the adolescents.

The study received approval from the Bioethics Committee at the University of Murcia (ID 2218/2018), the Ethics Committee of the Albacete University Hospital Complex, and the Albacete Integrated Care Management (ID 2021-85). All procedures were conducted in line with the Declaration of Helsinki, ensuring the protection of participants’ rights, including informed consent and confidentiality throughout the research process.

### 2.2. Involvement in Household Food Tasks

Adolescents’ participation in food-related household activities was measured with two items. The first asked how often they had helped prepare dinner during the previous seven days (past week), offering options from “never” to “seven times”. The second item assessed how many times they had helped with grocery shopping over the same period, using the same response categories [36].

### 2.3. Psychosocial Health

Psychosocial health was assessed via the self-report version of the Strengths and Difficulties Questionnaire (SDQ) [37], a widely utilized tool for assessing behavioral, emotional, and social issues in young individuals aged 11–17 years. This study employed the validated Spanish version of the SDQ (www.sdqinfo.org, accessed on 1 August 2021) [38]. The questionnaire included 25 items distributed across five subscales: (a) emotional problems, (b) conduct problems, (c) hyperactivity, (d) peer problems, and (e) prosocial behavior. Responses were recorded on a 3-point Likert scale: 0 = not true, 1 = somewhat true, and 2 = certainly true, with each subscale scoring from 0 to 10 points.

The total psychosocial problems score was derived from the first four subscales (emotional problems, conduct problems, hyperactivity, and peer problems), whereas the prosocial behavior subscale, which assessed positive social traits rather than difficulties, was analyzed separately. Additionally, two composite scores were calculated: the internalizing problems score (ranging from 0 to 20), which was obtained by summing the emotional and peer problems subscales, and the externalizing problems score (ranging from 0 to 20), which was derived from the sum of the conduct and hyperactivity subscales.

### 2.4. Covariates

Various covariates that could influence the relationship between participation in food-related tasks and adolescent psychosocial health were included in this study. The participants provided self-reported information on their age and sex. Socioeconomic status was assessed via the Family Affluence Scale (FAS-III), which considers six indicators of family wealth, including the number of rooms in the household, family vehicles, bathrooms, computers, trips taken in the past year, and dishwasher ownership, with a total score ranging from 0 to 13 [39].

Anthropometric measurements included body weight, which was recorded via a high-precision electronic scale (Tanita BC-545, Tokyo, Japan), and height, which was measured via a portable stadiometer (Leicester Tanita HR 001, Tokyo, Japan). Based on these data, body mass index (BMI) was calculated by dividing weight in kilograms by height in meters squared.

Energy intake was assessed via a self-administered food frequency questionnaire previously validated for the Spanish population [40]. To measure sleep duration, adolescents reported their usual bedtime and wake-up time on both weekdays and weekends. The average sleep duration was calculated via the following formula: [(weekday sleep duration × 5) + (weekend sleep duration × 2)] divided by 7 [41].

Physical activity and sedentary behavior were assessed with the Spanish version of the Youth Activity Profile (YAP-S), a validated questionnaire measuring activity levels in school, extracurricular, and leisure contexts [42].

### 2.5. Statistical Analysis

Data distribution was assessed using quantile–quantile plots, density graphs, and the Shapiro–Wilk test for normality. Categorical variables were reported as frequencies and percentages, while non-normally distributed continuous variables were summarized using medians and interquartile ranges (IQRs). Since no significant interaction was found between sex and involvement in food-related tasks in relation to SDQ scores (*p* > 0.05 in all cases), all subsequent analyses were conducted using the combined sample. An a priori power estimation was conducted to assess the adequacy of the sample size. Assuming a moderate effect size (*R*^2^ = 0.13), a significance level of α = 0.05, and a statistical power of 80%, a minimum of 114 participants would be required to detect meaningful associations in a generalized linear model (GLM) with nine covariates. Given that our final analytical sample included 634 adolescents, the study exceeds this threshold considerably, indicating sufficient power to detect the hypothesized effects. Primary analyses were conducted using GLMs with robust estimators (the SMDM method) and a Gaussian distribution to account for heteroscedasticity and outliers. The results were expressed as unstandardized beta coefficients (*B*) with 95% confidence intervals (CIs). All models were adjusted for age, sex, socioeconomic status, BMI, sleep duration, physical activity, sedentary behavior, and energy intake. Additionally, to address the potential impact of missing data, we conducted multiple imputation by chained equations (MICE) via the ‘mice’ package (v3.16.0). Following established recommendations, 40 imputed datasets were generated, exceeding 100 times the maximum percentage of missingness in any given variable. To assess the plausibility of the missing at random (MAR) assumption, Little’s missing completely at random (MCAR) test was performed using the ‘mcar_test’ function from the ‘naniar’ package (v0.7.0). The test indicated that the data were not missing completely at random (MCAR) (chi-square [*χ*^2^] = 909, degrees of freedom [*df*] = 275, *p* < 0.001). However, descriptive comparisons between participants with and without missing data on key variables revealed no substantial differences (Appendix A), supporting the plausibility of the MAR assumption and the validity of the imputation approach. All statistical analyses were conducted using R software (v4.4.0; R Core Team, Vienna, Austria) and RStudio (v2024.04.1+748; Posit PBC, Boston, MA, USA), with a significance level set at *p* < 0.05.

## 3. Results

Table 1 summarizes the characteristics of the study participants (*n* = 634). The median age was 14.0 years (IQR 13.0 to 15.0), with 43% boys (*n* = 273) and 57% girls (*n* = 361). The median BMI was 21.7 kg/m^2^ (IQR 19.3 to 25.1), and the median FAS-III score was 8.0 (IQR 7.0 to 9.0), indicating a medium socioeconomic level. Participants reported a median sleep duration of 501.4 min per night (IQR 458.6 to 531.4), and the median energy intake was 2609.4 kcal/day (IQR 1964.7 to 3468.9). The median physical activity score was 2.6 (IQR 2.2 to 3.1), and sedentary behavior had a similar score of 2.6 (IQR 2.2 to 3.0). Regarding food-related tasks, 51% of adolescents reported helping to prepare dinner five to six times in the past week, while 23% did so every day. In contrast, only 0.9% never helped with dinner preparation. Similarly, 43% helped with grocery shopping five to six times, and 21% did so seven times; only 0.9% never participated in this task. In terms of psychosocial health outcomes, the median scores were as follows: emotional problems 3.0 (IQR 1.0 to 5.0), conduct problems 2.0 (IQR 1.0 to 3.0), hyperactivity 4.0 (IQR 3.0 to 6.0), and peer problems 2.0 (IQR 1.0 to 3.0). The prosocial behavior score had a median of 8.0 (IQR 7.0 to 9.0), while externalizing and internalizing problems scored 6.0 (IQR 4.0 to 8.0) and 5.0 (IQR 2.0 to 8.0), respectively. The total difficulties score had a median of 11.0 (IQR 7.0 to 16.0).

Table 2 presents the characteristics of the study participants according to the frequency of helping to prepare dinner. Adolescents who reported helping with dinner preparation seven times per week showed a slightly higher median physical activity score (2.9; IQR 2.5 to 3.2) compared to those who helped less frequently. Median energy intake was also higher among those most frequently involved (2931.9 kcal/day; IQR 2206.7 to 3661.4). In terms of psychosocial variables, those with higher involvement in dinner preparation tended to report lower levels of emotional problems (median = 2.0; IQR 1.0 to 4.0), conduct problems (median = 2.0; IQR 1.0 to 3.0), and internalizing problems (median = 4.0; IQR 2.0 to 7.0), as well as higher prosocial behavior scores (median = 9.0; IQR 8.0 to 10.0), compared to their peers with less involvement. The total-difficulties score was lower among adolescents who helped every day (median = 9.0; IQR 6.0 to 14.0), suggesting more favorable psychosocial profiles.

Table 3 shows participant characteristics according to the frequency of helping with grocery shopping. Adolescents who reported shopping seven times per week had a higher median energy intake (2914.6 kcal/day; IQR 2095.6 to 3577.3) and physical activity score (2.8; IQR 2.4 to 3.2) than those with lower involvement. These participants also had lower median scores for emotional problems (2.0; IQR 1.0 to 4.0), conduct problems (2.0; IQR 1.0 to 3.0), and internalizing problems (4.0; IQR 2.0 to 7.0), and higher prosocial behavior scores (median = 9.0; IQR 8.0 to 10.0). As with dinner preparation, greater involvement in grocery shopping was associated with lower total-difficulties scores (median = 10.0; IQR 6.0 to 14.0), indicating better psychosocial functioning.

Figure 1 presents the adjusted marginal means of psychosocial health scores according to the frequency of helping to prepare food for dinner. An inverse association was observed between food preparation involvement and several psychosocial problems. Adolescents who helped seven times per week reported significantly lower scores in conduct problems (*B* = −2.00; 95% CI −3.30 to −0.69; *p* = 0.003), peer problems (*B* = −2.83; 95% CI −4.29 to −1.38; *p* < 0.001), internalizing problems (*B* = −3.90; 95% CI −7.03 to −0.77; *p* = 0.015), and total psychosocial difficulties (*B* = −5.74; 95% CI −10.68 to −0.80; *p* = 0.023), compared to those who never helped. Conversely, those who helped seven times per week had higher prosocial behavior than their counterparts who never helped (*B* = 1.69; 95% CI: 0.14 to 3.24; *p* = 0.033). The full results are detailed in Appendix A.

Figure 2 displays the adjusted marginal means of psychosocial health domains based on the frequency of helping to shop for food. Similar patterns were found, with lower conduct problems (*B* = −2.11; 95% CI −3.42 to −0.81; *p* = 0.002), peer problems (*B* = −2.88; 95% CI −4.34 to −1.42; *p* < 0.001), internalizing problems (*B* = −4.16; 95% CI −7.28 to −1.04; *p* = 0.009), and total psychosocial difficulties (*B* = −6.31; 95% CI −11.24 to −1.39; *p* = 0.012) associated with more frequent involvement, especially among those who helped five or more times per week. Conversely, adolescents who helped seven times per week had higher prosocial behavior than their peers who never helped (*B* = 1.56; 95% CI: 0.01 to 3.11; *p* = 0.049). Complete regression models are provided in Appendix A.

## 4. Discussion

This study found that greater involvement in food-related household tasks was associated with more favorable psychosocial health among Spanish adolescents. Specifically, a higher frequency of participation in both meal preparation and grocery shopping was linked to lower levels of conduct problems, peer problems, internalizing difficulties, total difficulties, along with higher prosocial behavior [43]. These results remained significant even after adjusting for relevant sociodemographic, lifestyle, and nutritional covariates, reinforcing the potential importance of everyday routines in adolescent mental well-being. These findings are consistent with prior research suggesting that structured routines contribute positively to emotional and social development during adolescence [21,44]. However, it is important to consider that the directionality of these associations cannot be established given the cross-sectional nature of the study. In fact, it is possible that adolescents with impaired psychosocial health are less inclined to participate in household food-related tasks, rather than the reverse. This possibility of reverse causation should be kept in mind when interpreting the results.

Adolescents who reported helping with grocery shopping reported behavioral difficulties. This activity may represent a valuable opportunity for adolescents to build executive function skills, such as planning, decision making, and evaluating options [45]. In addition, grocery shopping may promote real-world interactions and problem solving, helping to boost confidence and social engagement [46,47]. Adolescents may benefit from learning these food management tasks early on, even if they see them as less relevant at their current stage of life. Integrating basic food skills into school programs could help strengthen their food literacy and independence [48]. It is worth noting that some adolescents reported participating in grocery shopping seven times per week, which may reflect family routines involving frequent small purchases, cultural shopping practices, or social desirability bias. This possibility should be considered when interpreting these findings.

Regarding meal preparation, consistent participation was associated with lower behavioral problems. Cooking activities may help strengthen emotional regulation, patience, and collaboration, especially with family [49,50]. Previous studies have shown that adolescents who engage in cooking are more likely to report improved self-efficacy and a sense of responsibility—factors strongly linked to mental health [12,15,21]. Additional evidence highlights that involvement in cooking may support adolescent development by fostering independence and emotional skills [51]. Cooking also fosters meaningful family interactions, which can act as a protective buffer against psychosocial stressors [16,19,45]. These findings likely reflect the cumulative benefits of increased autonomy, executive functioning, and family interactions associated with regular participation in food-related tasks.

Understanding these findings has practical implications for both families and public health professionals. It is also worth highlighting that encouraging any level of contribution to food-related tasks could be more meaningful than focusing solely on frequency or intensity of participation. Our results suggest that the most relevant distinction lies between no contribution and any contribution, rather than between increasing levels of involvement. These benefits are supported by longitudinal research demonstrating that adolescents’ participation in household chores (even light, occasional tasks) can lead to improvements in emotional well-being [52,53]. For example, a longitudinal study in the U.S. involving European American families aged 8–18 years found that participation in housework was associated with better emotional adjustment and prosocial behavior over time [52]. Similarly, a prospective cohort study from the UK found that increasing light daily activities (including household chores) during adolescence was associated with a reduced risk of depressive symptoms by age 18 [53]. Encouraging adolescents to participate in food-related tasks could represent a simple and accessible way to enhance well-being and foster personal growth [18]. Additionally, specific programmatic strategies could be considered, such as integrating life skills modules into school curricula that combine cooking, budgeting, and meal planning with mental health education [54]. Meta-analytical evidence shows that such life skill programs substantially improve adolescents’ emotional regulation, stress resilience, and mental health outcomes in low- and middle-income settings [55]. An intensive cooking intervention from New Zealand demonstrated long-term increases in cooking self-efficacy and mental well-being among adolescents aged 12–15 [56]. These activities may contribute to the development of emotional resilience, social skills, and a sense of belonging, as suggested by previous findings [57,58,59]. Integrating them into broader school- or family-based mental health initiatives could enhance their impact on adolescent well-being. Such strategies may be particularly valuable in supporting adolescent mental health amidst increasingly fragmented daily routines and growing digital and socioeconomic challenges [25,58,60].

This study has several limitations that should be acknowledged. First, its cross-sectional design does not allow for the establishment of causal relationships, and future longitudinal studies are essential to verify the directionality of these associations and to assess whether promoting food-related responsibilities could serve as a sustainable strategy to enhance adolescent psychosocial health. Second, a large proportion of the initial sample was excluded due to missing data, particularly regarding psychosocial health outcomes, which could potentially introduce selection bias. Although no formal statistical tests were applied, visual inspection of descriptive characteristics between the included and excluded participants revealed general similarities in age, sex, socioeconomic status, BMI, and sleep duration, suggesting that the analytical sample was reasonably representative of the broader cohort. Furthermore, results from Little’s MCAR test indicated that the data were not missing completely at random. However, the descriptive comparisons support the plausibility of the Missing at Random (MAR) assumption, justifying the use of multiple imputation by chained equations. The consistency of the results across imputed and complete case datasets strengthens the reliability of the conclusions. Third, we did not assess certain potential confounders, such as personality traits or the quality of family relationships, which may influence both household task involvement and psychosocial outcomes. Fourth, the use of self-reported measures may be subject to recall bias and social desirability bias. In addition, although our questions were adapted from Larson et al. [36], no formal validation study of these items was conducted in our population, which may limit the precision of exposure measurement. Fifth, the findings are based on a specific regional sample in Spain, which may limit generalizability. Finally, the study was conducted during the academic year immediately following the Coronavirus Disease 2019 (COVID-19) pandemic, and it is possible that increased time spent at home affected adolescents’ involvement in household responsibilities and their psychosocial well-being. Nonetheless, the study is strengthened by the use of validated instruments, a large sample size, and comprehensive adjustment for confounding variables. Furthermore, it is based on a relatively large sample of adolescents, which, according to a power estimation, provides sufficient statistical power to detect meaningful associations in multivariable models.

## 5. Conclusions

Although adolescent psychosocial health is influenced by multiple factors, our findings suggest that regular involvement in food-related household tasks may serve as a protective factor against conduct problems, peer problems, internalizing problems, and total difficulties, while also enhancing prosocial behavior. However, given the cross-sectional design, conclusions regarding causality should be made cautiously, and further longitudinal research is needed to confirm these associations and assess their long-term impact. These results highlight the relevance of daily structured routines, such as meal preparation and grocery shopping, as a potential support for mental well-being during adolescence. Rather than prescribing specific interventions, our findings suggest that creating opportunities for adolescents to engage in household food tasks could form a part of broader, accessible, and family-centered strategies to promote emotional resilience, improve family interactions, and foster life skills. Future public health interventions and educational programs might benefit from integrating these everyday practices as part of broader efforts to support adolescent development and mental health.

## Figures and Tables

**Figure 1 nutrients-17-02273-f001:**
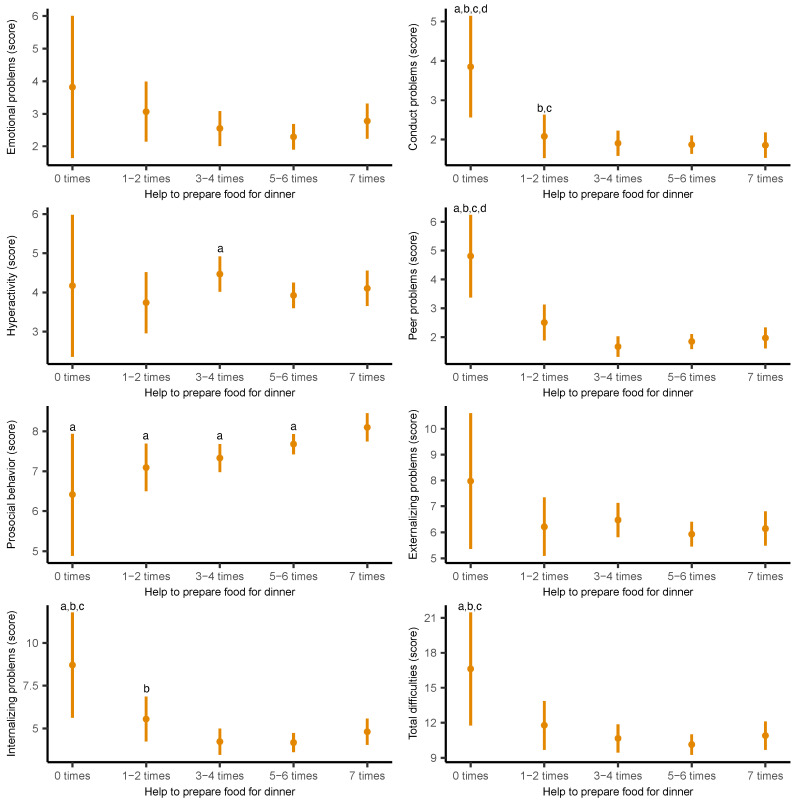
Adjusted marginal means of the individual and overall domains of the Strengths and Difficulties Questionnaire by frequency of adolescents’ helping to prepare food for dinner. Means were estimated from generalized linear models adjusted for age, sex, socioeconomic status, body mass index, sleep duration, physical activity, sedentary behavior, and energy intake. ^a^ Significant differences from “seven times” (*p* < 0.05); ^b^ significant differences from “five or six times” (*p* < 0.05); ^c^ significant differences from “three or four times” (*p* < 0.05); ^d^ significant differences from “one or two times” (*p* < 0.05).

**Figure 2 nutrients-17-02273-f002:**
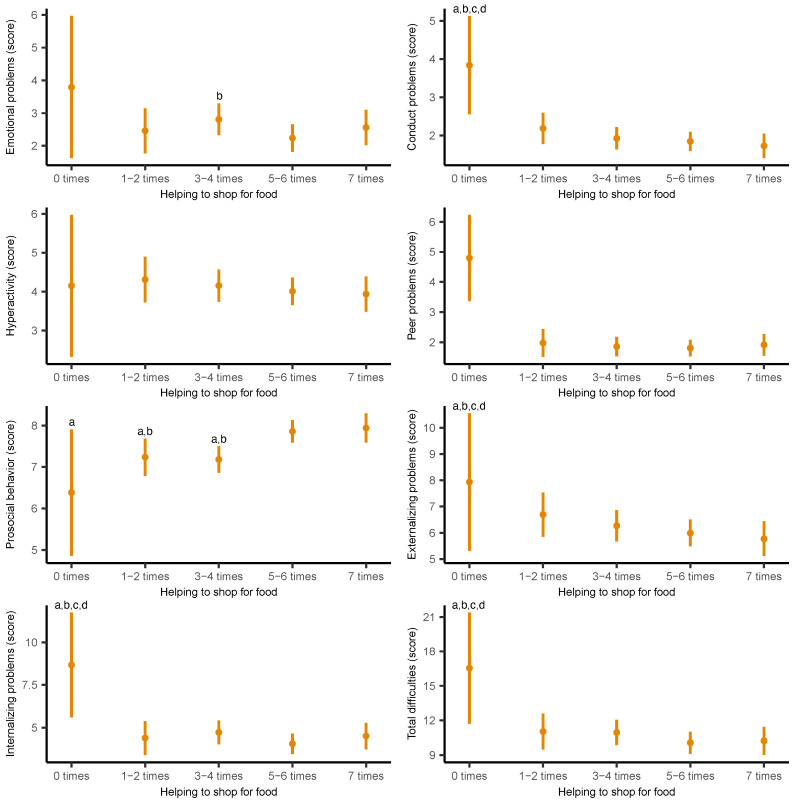
Adjusted marginal means of the individual and overall domains of the Strengths and Difficulties Questionnaire by frequency of adolescents’ helping to shop for food. Means were estimated from generalized linear models adjusted for age, sex, socioeconomic status, body mass index, sleep duration, physical activity, sedentary behavior, and energy intake. ^a^ Significant differences from “seven times” (*p* < 0.05); ^b^ significant differences from “five or six times” (*p* < 0.05); ^c^ significant differences from “three or four times” (*p* < 0.05); ^d^ significant differences from “one or two times” (*p* < 0.05).

**Table 1 nutrients-17-02273-t001:** Descriptive characteristics of the sample (*N* = 634).

Variable	*N* = 634 ^1^
Age (years)	14.0 (13.0, 15.0)
Sex	
Boys	273 (43.0%)
Girls	361 (57.0%)
FAS-III (score)	8.0 (7.0, 9.0)
Overall sleep duration (minutes)	501 (459, 531)
YAP-S physical activity (score)	2.6 (2.2, 3.1)
YAP-S sedentary behaviors (score)	2.6 (2.2, 3.0)
Energy intake (kcal)	2609 (1965, 3469)
BMI (kg/m^2^)	21.7 (19.3, 25.1)
Helping to prepare food for dinner	
Never	6 (0.9%)
One or two times	36 (5.7%)
Three or four times	120 (19.0%)
Five or six times	325 (51.0%)
Seven times	147 (23.0%)
Helping to shop for food	
Never	6 (0.9%)
One or two times	67 (11.0%)
Three or four times	154 (24.0%)
Five or six times	274 (43.0%)
Seven times	133 (21.0%)
Emotional problems (SDQ score)	3.0 (1.0, 5.0)
Conduct problems (SDQ score)	2.0 (1.0, 3.0)
Hyperactivity (SDQ score)	4.0 (3.0, 6.0)
Peer problems (SDQ score)	2.0 (1.0, 3.0)
Prosocial behavior (SDQ score)	8.0 (7.0, 9.0)
Externalizing problems (SDQ score)	6.0 (4.0, 8.0)
Internalizing problems (SDQ score)	5.0 (2.0, 8.0)
Total difficulties (SDQ score)	11.0 (7.0, 16.0)

^1^ Median (interquartile range) or number (percentage). BMI, body mass index; FAS-III, Family Affluence Scale-III; YAP-S, Spanish Youth Activity Profile; SDQ, Strengths and Difficulties Questionnaire.

**Table 2 nutrients-17-02273-t002:** Descriptive characteristics of the sample based on helping to prepare food for dinner frequency.

Variable	Never (*n* = 6)	One or Two Times (*n* = 36)	Three or Four Times (*n* = 120)	Five or Six Times (*n* = 325)	Seven Times (*n* = 147)
Age (years)	13.5 (13.0, 15.0)	13.0 (12.0, 14.0)	14.0 (13.0, 15.0)	14.0 (13.0, 15.0)	14.0 (12.0, 15.0)
Sex					
Boys	3.0 (50.0%)	20.0 (55.6%)	59.0 (49.2%)	142.0 (43.7%)	49.0 (33.3%)
Girls	3.0 (50.0%)	16.0 (44.4%)	61.0 (50.8%)	183.0 (56.3%)	98.0 (66.7%)
FAS-III (score)	8.0 (6.0, 8.0)	8.0 (7.0, 9.0)	8.0 (7.0, 9.0)	8.0 (7.0, 10.0)	8.0 (7.0, 10.0)
Overall sleep duration (minutes)	407.1 (407.1, 522.9)	503.6 (458.6, 537.9)	492.9 (450.0, 518.6)	501.4 (458.6, 531.4)	505.7 (467.1, 535.7)
YAP-S physical activity (score)	2.6 (2.4, 2.9)	2.5 (2.0, 3.1)	2.6 (2.1, 2.9)	2.6 (2.2, 3.1)	2.7 (2.3, 3.1)
YAP-S sedentary behaviors (score)	2.5 (1.8, 3.6)	2.6 (2.2, 3.2)	2.6 (2.1, 3.0)	2.6 (2.2, 3.0)	2.4 (2.0, 2.8)
Energy intake (kcal)	5556.1 (3752.0, 8205.6)	2614.4 (1777.0, 3662.4)	2581.8 (1977.7, 3316.8)	2589.3 (1973.1, 3400.9)	2694.1 (1887.9, 3788.5)
BMI (kg/m^2^)	21.9 (18.8, 22.6)	21.1 (18.3, 24.2)	21.8 (19.5, 25.1)	21.6 (19.0, 24.8)	22.4 (19.6, 26.8)
Helping to shop for food					
Never	6.0 (100.0%)	0.0 (0.0%)	0.0 (0.0%)	0.0 (0.0%)	0.0 (0.0%)
One or two times	0.0 (0.0%)	13.0 (36.1%)	21.0 (17.5%)	25.0 (7.7%)	8.0 (5.4%)
Three or four times	0.0 (0.0%)	10.0 (27.8%)	55.0 (45.8%)	78.0 (24.0%)	11.0 (7.5%)
Five or six times	0.0 (0.0%)	10.0 (27.8%)	32.0 (26.7%)	182.0 (56.0%)	50.0 (34.0%)
Seven times	0.0 (0.0%)	3.0 (8.3%)	12.0 (10.0%)	40.0 (12.3%)	78.0 (53.1%)
Emotional problems (SDQ score)	5.5 (3.0, 7.0)	3.0 (1.0, 6.0)	3.0 (1.0, 5.0)	3.0 (1.0, 5.0)	4.0 (2.0, 6.0)
Conduct problems (SDQ score)	5.0 (2.0, 5.0)	2.0 (0.5, 3.0)	2.0 (1.0, 3.0)	1.0 (1.0, 3.0)	2.0 (1.0, 3.0)
Hyperactivity (SDQ score)	5.0 (3.0, 6.0)	4.0 (2.5, 5.0)	5.0 (3.0, 6.0)	4.0 (3.0, 5.0)	4.0 (3.0, 6.0)
Peer problems (SDQ score)	5.5 (4.0, 6.0)	2.0 (1.0, 4.5)	2.0 (1.0, 3.0)	2.0 (1.0, 3.0)	2.0 (1.0, 3.0)
Prosocial behavior (SDQ score)	6.0 (5.0, 8.0)	8.0 (6.0, 9.0)	8.0 (6.0, 9.0)	8.0 (7.0, 9.0)	9.0 (7.0, 10.0)
Externalizing problems (SDQ score)	9.0 (6.0, 11.0)	6.0 (4.0, 10.0)	7.0 (4.0, 9.0)	6.0 (4.0, 8.0)	6.0 (4.0, 9.0)
Internalizing problems (score)	10.5 (9.0, 13.0)	6.0 (2.0, 10.0)	5.0 (2.0, 8.0)	5.0 (2.0, 8.0)	6.0 (3.0, 9.0)
Total difficulties (SDQ score)	19.5 (14.0, 24.0)	12.0 (8.0, 15.0)	12.0 (7.0, 15.5)	11.0 (7.0, 15.0)	12.0 (8.0, 17.0)

Median (interquartile range) or number (percentage). BMI, body mass index; FAS-III, Family Affluence Scale-III; YAP-S, Spanish Youth Activity Profile; SDQ, Strengths and Difficulties Questionnaire.

**Table 3 nutrients-17-02273-t003:** Descriptive characteristics of the sample based on helping to shop for food frequency.

Variable	Never (*n* = 6)	One or Two Times (*n* = 67)	Three or Four Times (*n* = 154)	Five or Six Times (*n* = 274)	Seven Times (*n* = 133)
Age (years)	13.5 (13.0, 15.0)	14.0 (13.0, 15.0)	14.0 (13.0, 16.0)	14.0 (13.0, 15.0)	14.0 (12.0, 15.0)
Sex					
Boys	3.0 (50.0%)	35.0 (52.2%)	73.0 (47.4%)	111.0 (40.5%)	51.0 (38.3%)
Girls	3.0 (50.0%)	32.0 (47.8%)	81.0 (52.6%)	163.0 (59.5%)	82.0 (61.7%)
FAS-III (score)	8.0 (6.0, 8.0)	9.0 (7.0, 9.0)	8.0 (7.0, 9.0)	8.0 (7.0, 9.0)	8.0 (6.0, 9.0)
Overall sleep duration (minutes)	407.1 (407.1, 522.9)	492.9 (454.3, 518.6)	492.9 (454.3, 522.9)	505.7 (467.1, 540.0)	497.1 (454.3, 531.4)
YAP-S physical activity (score)	2.6 (2.4, 2.9)	2.5 (2.0, 3.0)	2.6 (2.2, 2.9)	2.6 (2.2, 3.1)	2.7 (2.3, 3.2)
YAP-S sedentary behaviors (score)	2.5 (1.8, 3.6)	2.8 (2.4, 3.4)	2.6 (2.2, 3.0)	2.4 (2.0, 3.0)	2.4 (2.0, 2.8)
Energy intake (kcal)	5556.1 (3752.0, 8205.6)	2831.4 (1882.0, 3509.8)	2507.2 (1954.7, 3301.6)	2583.9 (1994.5, 3410.5)	2788.9 (1936.1, 3548.1)
BMI (kg/m^2^)	21.9 (18.8, 22.6)	21.2 (19.6, 24.1)	22.5 (19.1, 25.6)	21.3 (19.1, 24.8)	21.8 (19.6, 26.4)
Helping to prepare food for dinner					
Never	6.0 (100.0%)	0.0 (0.0%)	0.0 (0.0%)	0.0 (0.0%)	0.0 (0.0%)
One or two times	0.0 (0.0%)	13.0 (19.4%)	10.0 (6.5%)	10.0 (3.6%)	3.0 (2.3%)
Three or four times	0.0 (0.0%)	21.0 (31.3%)	55.0 (35.7%)	32.0 (11.7%)	12.0 (9.0%)
Five or six times	0.0 (0.0%)	25.0 (37.3%)	78.0 (50.6%)	182.0 (66.4%)	40.0 (30.1%)
Seven times	0.0 (0.0%)	8.0 (11.9%)	11.0 (7.1%)	50.0 (18.2%)	78.0 (58.6%)
Emotional problems (SDQ score)	5.5 (3.0, 7.0)	3.0 (1.0, 5.0)	3.5 (1.0, 6.0)	3.0 (1.0, 5.0)	3.0 (1.0, 6.0)
Conduct problems SDQ (score)	5.0 (2.0, 5.0)	2.0 (1.0, 4.0)	2.0 (1.0, 3.0)	1.0 (1.0, 3.0)	1.0 (1.0, 2.0)
Hyperactivity (SDQ score)	5.0 (3.0, 6.0)	5.0 (2.0, 6.0)	4.0 (3.0, 6.0)	4.0 (3.0, 6.0)	4.0 (3.0, 5.0)
Peer problems (SDQ score)	5.5 (4.0, 6.0)	2.0 (1.0, 4.0)	2.0 (1.0, 3.0)	2.0 (1.0, 3.0)	2.0 (1.0, 4.0)
Prosocial behavior (SDQ score)	6.0 (5.0, 8.0)	8.0 (6.0, 9.0)	8.0 (6.0, 9.0)	8.0 (7.0, 9.0)	9.0 (7.0, 10.0)
Externalizing problems (SDQ score)	9.0 (6.0, 11.0)	7.0 (4.0, 10.0)	6.0 (4.0, 8.0)	6.0 (4.0, 8.0)	6.0 (4.0, 8.0)
Internalizing problems (SDQ score)	10.5 (9.0, 13.0)	5.0 (2.0, 8.0)	6.0 (3.0, 8.0)	4.5 (2.0, 8.0)	6.0 (3.0, 9.0)
Total difficulties (SDQ score)	19.5 (14.0, 24.0)	11.0 (8.0, 16.0)	11.5 (8.0, 16.0)	11.0 (7.0, 15.0)	11.0 (7.0, 16.0)

Median (interquartile range) or number (percentage). BMI, body mass index; FAS-III, Family Affluence Scale-III; YAP-S, Spanish Youth Activity Profile; SDQ, Strengths and Difficulties Questionnaire.

## Data Availability

The datasets generated and analyzed during this research are available from the corresponding author upon reasonable request. Public sharing of the data is restricted due to the involvement of minors.

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
