# Peer review of "Is Involvement in Food Tasks Associated with Psychosocial Health in Adolescents? The EHDLA Study"

_nutrients, 2025, doi:10.3390/nu17142273_

Round 1
Reviewer 1 Report
Comments and Suggestions for Authors
I believe the Introduction section should be expanded to include three additional themes that are highly relevant but currently missing.
1., there should be a discussion on gender differences in domestic responsibilities and how these might influence both the frequency of food-related task involvement and psychosocial outcomes in adolescents. Gender norms can affect not only participation but also the meaning adolescents assign to these tasks.
2., the authors should touch on cultural and familial eating practices, especially in Spain, where mealtime and food preparation often hold strong social and emotional significance. This would add depth to the study’s context.
3., digital and media influences on food behavior in adolescents could be mentioned. Many young people are influenced by social media trends, cooking tutorials, and online content related to food, which may impact both their interest in participating in food tasks and their mental well-being.
The authors state that the study was conducted in line with the Helsinki Declaration. This is good ethical practice, but it is important to clarify whether the research also followed the Publication Manual of the American Psychological Association (APA) Code of Ethics, especially in terms of confidentiality, informed consent from minors, and responsible data handling. This clarification would strengthen the ethical transparency of the paper.
Table 1 requires editing for clarity and readability.
On a positive note, I would like to commend the authors for their use of recent and high-quality references. The sources are relevant, well-chosen, and support the rationale and discussion effectively. This helps situate the study within the current body of research and shows strong scholarly grounding.
Author Response
Quito, 20th June 2025
Dear Editors,
Nutrients
Re: Decision on Manuscript ID nutrients-3704606, entitled “Is involvement in food tasks associated with psychosocial health in adolescents? The EHDLA Study”
We are pleased to submit a revised version of our manuscript for your consideration. We greatly appreciate the constructive feedback from the reviewers, which has significantly improved the quality of our work.
Please find attached a detailed, point-by-point response to each of the reviewers’ comments. We have highlighted all changes within the revised manuscript for ease of review.
We hope that the revised manuscript meets the expectations of the editorial team and reviewers.
We sincerely thank you for the opportunity to contribute to Nutrients.
Yours sincerely

Reviewer 2 Report
Comments and Suggestions for Authors
The authors present a cross-sectional study on the association of food-related household activities with measures of psychosocial health.
The overall rationale of the study is clear.
Introduction:
L. 85: Please clarify, whether previous evidence is observational or interventional in nature.
Methods:
L. 103: A major part of the initial cohort was excluded due to missing data. Please clarify, whether this introduces bias, as the excluded part might be special with respect to low household contribution (as part of overall lack of contribution and engagement).
Results:
A considerable fraction of the subjects indicated, that they helped seven times per week with grocery shopping. This implies daily grocery shopping - even on sundays. Is that plausible?
Fig. 1 and 2 have two separate legends; this is unusual. Please merge or transform one legend into main text.
Fig. 1 is incomprehensible, as axis' legends and titles are missing.
The supplemental material was not attached, thus could not be checked.
Discussion:
Although the limitations section indicates, that this study cannot provide insight into causality, the actual consequence of that learning is missing: causality could be opposite to what is implied. Impaired psychosocial health leads to low household participation, not the other way round. Depression is typically connected to low physical activity, low interest in social interaction, low interest in complex planning behavior (such as food preparation, groceries). Ergo, one cannot simply assume, that enforced activity ameliorates psychosocial health problems.
Furthermore: The only cohort subgroup with considerably distinct psychosocial properties is the one with zero contribution (to food preparation and/or groceries). The other four are virtualy indistinguishable. Ergo, the notion cannot be to encourage "more contribution", but - if only (assuming causality) - to encourage "any contribution at all".
Author Response

(The authors gave the same response as above.)

Reviewer 3 Report
Comments and Suggestions for Authors
Dear corresponding Author, thank you for submitting your work to Nutrients journal and congratulations for the research.
Brief Summary
The study presents a cross-sectional analysis conducted on 634 Spanish adolescents (12-17 years) from Murcia Region to examine the association between involvement in food-related household tasks (meal preparation and grocery shopping) and psychosocial health. Participants reported weekly frequency of participation in these tasks through self-administered questionnaires, while psychosocial health was assessed through the Strengths and Difficulties Questionnaire (SDQ).
General Comments
The topic addressed is certainly interesting and of practical relevance, however the study presents several methodological limitations that compromise the robustness of conclusions. For example, what distinguishes the 634 included participants from the 504 excluded ones? This information is crucial for evaluating possible selection bias. The methodology presents some problematic aspects. No statistical power analysis is reported, making difficult to evaluate if the sample size is adequate to detect the observed effects. Moreover, the exclusive use of self-reported measures introduces potential recall and social desirability bias, particularly relevant when studying adolescents.
Specific Comments
- Line 102-104: The exclusion of 504 participants (44.3%) for "missing information on key variables" is problematic and requires more details. What were exactly these key variables? How were distributed the characteristics of excluded participants compared to included ones? This analysis is fundamental for evaluating the representativeness of final sample. Can you clarify better this aspect?
- Lines 114-119: The measurement of expositive variables appears too simplified. The questions about food tasks would have needed a more rigorous validation. Furthermore, the exact reference period is not specified ("past week") - is it about a typical week or the week immediately preceding the administration? Also on this point can you clarify better?
- Line 123: The reference to the website for Spanish version of SDQ is not appropriate for a scientific journal. It should be substituted with the appropriate citation of Spanish validation of the instrument, does it exist?
- Results Section: It lacks a descriptive analysis that compares participants' characteristics based on level of involvement in food tasks. This would be usefull to understand better the studied population.
- Figures 1 and 2: The figures are informatively useful but need more detailed captions. What exactly represent the letters (a, b, c, d) in the graphs? This should be clarified in the legend. So they are objectively not understandable. Furthermore there are two descriptions of "Figure 2" at line 211 and 222, it's not clear and creates confusion.
- Lines 227-235: The discussion starts appropriately summarizing main results, but the connection with existing literature could be more critical. Authors tend to interpret results in a rather optimistic way without considering adequately alternative explanations.
- Lines 265-272: The limitations section is too brief and superficial. It should include discussions on: selection bias due to high exclusion rate, potential unmeasured confounders (e.g. personality, family relationships), limitations of self-reported measures, and possibility of reverse causation. The limitations of this study are notable, it's necessary to clarify them.
- CRITICAL PROBLEM: The article completely lacks the "Author Contributions" section, which is a mandatory requirement for MDPI publications. This section must be added before any further consideration for publication. I would also like to know what role the authors had...
The conclusion tends to be too prescriptive considering the methodological limitations of study. Phrases like "Encouraging participation in daily food routines may represent a practical, low-cost approach" go beyond what the data can support in a cross-sectional design.
Overall, while the topic is interesting and results suggestive, the study requires substantial revisions to improve methodological rigor and appropriateness of drawn conclusions. I wait for an improved version to read.
Author Response

(The authors gave the same response as above.)

Reviewer 4 Report
Comments and Suggestions for Authors
This manuscript is methodologically sound, clearly written, and addresses an important social issue from a practical perspective. The study explores a relatively underexamined area, the association between adolescents’ involvement in food-related household tasks and their psychosocial health, contributing valuable new insights. The results are consistent with existing international literature while also raising novel implications, particularly for educational and developmental practices. The discussion is generally strong, but several additions could further enhance the manuscript:
The current suggestions for implementation are somewhat general. It would strengthen the practical relevance of the study to provide specific, real-world program examples. For instance, the authors could propose the integration of a "Life Skills" module into school curricula, combining cooking, budgeting, and meal planning with mental health education. Similar programs have been successfully implemented in countries such as Canada and New Zealand and have been shown to support adolescents’ independence and emotional development.
Given that the study was conducted during the 2021/22 academic year, it would be appropriate to briefly acknowledge the potential impact of the COVID pandemic. Increased time spent at home during lockdowns may have influenced adolescents’ participation in household tasks, as well as their mental health status. A short note such as the following could be added - Since the study took place in the academic year immediately following the COVID-19 pandemic, it is possible that increased time at home affected adolescents’ involvement in household responsibilities and their psychosocial well-being -
In summary, the manuscript is suitable for publication and would benefit from these minor, content-related enhancements to further increase its impact and contextual relevance.
Author Response

(The authors gave the same response as above.)

Round 2
Reviewer 1 Report
Comments and Suggestions for Authors
Thank you for your improvements. The article is much better now.
Author Response
Ok Thank you very much

Reviewer 2 Report
Comments and Suggestions for Authors
The authors have improved the manuscript in accordance to the reviewer's suggestions.
Please adjust the abstract accordingly, in order to reflect the more balanced discussion / conclusion of the main text.
Author Response
OK thank you very much

Reviewer 3 Report
Comments and Suggestions for Authors
I have read the authors’ response to my comments. I believe that the work still presents some critical issues, but with these revisions it reaches an adequate level of consistency to be published in its current form.
Author Response
OK thank you very much
